# Chemical Durability of Thermal Insulating Materials in Hall-Héroult Electrolysis Cells

**Raymond Luneng** [1]**, Søren N. Bertel** [2]**, Jørgen Mikkelsen** [2] **, Arne Petter Ratvik** [3]
**and Tor Grande** [1,*]

1. Department of Materials Science and Engineering, NTNU Norwegian University of Science and Technology, NO-7491 Trondheim, Norway
2. Skamol A/S, Sletvej 2C, DK-8310 Tranbjerg, Denmark
3. SINTEF Industry, NO-7465 Trondheim, Norway
* Correspondence: tor.grande@ntnu.no

**Abstract:** The most common thermal insulating materials used in the cathode lining in aluminum electrolysis cells are Moler (diatomaceous earth), calcium silicate, or vermiculite based materials. The thermal insulation layer is critical for the overall thermal stability of the cell and is vulnerable to volatile species, such as sodium vapor, that may penetrate through the carbon cathode and refractory layer. Here, we present an investigation of the chemical degradation of typical thermal insulating materials by exposure to sodium vapor in a laboratory test. Changes in microstructure and chemical and mineralogical composition of the exposed materials were characterized by electronic microscopy and powder X-ray diffraction. The materials possess different reaction patterns, ranging from deformation by creep to formation of a glassy layer reducing further sodium penetration. The results from the laboratory test were compared with chemical reactions with sodium predicted by computational thermodynamics and discussed with respect to relevant ternary phase diagrams.

**Keywords:** thermal insulating materials; aluminum electrolysis cell; sodium vapor; Moler; calcium silicate; vermiculite; degradation

---

## 1. Introduction

The production of aluminum by molten salt electrolysis is energy demanding with a considerable amount of input energy lost in the form of heat release to the surroundings. Today, state-of-the-art smelters are able to reach specific energy consumptions as low as 12 kWh/kg Al, while potline amperages have increased up to 500–600 kA [1,2]. The continued move towards low energy cells (lean cells) will require better insulated cells, as less heat will be generated to maintain operational temperature. This will also increase the heat gradient towards the bottom insulation layer. Further, the cell lifetime is limited, in most cases, by carbon cathode wear, which has become a major challenge as the industry has progressed towards high amperage cells [3,4]. One of the most important factors for the increase in the potline amperages was the shift from anthracitic carbon to graphitized carbon in the cathode blocks. While this shift ensured lower energy consumption through the increased electrical conductivity of the cathode, the wear resistance of the cathode blocks decreased, hence negatively affecting the cell lifetime [4]. Increased lifetime may be achieved by increasing the carbon cathode thickness at the expense of the refractory layer. However, this will also put higher demand on the thermal insulating layer and may increase the exposure to volatile species such as Na and $NaAlF_4$. Sodium vapor, which most likely [5,6] is the first chemical species diffusing through the refractory layer, can react with the highly porous insulation material and degrade the properties. While the stability of the thermal insulation materials has not received extensive attention in the literature [7–13],

the forecast to reduce the thickness of the refractory layer actuates the understanding of the chemical and thermal stability of these materials.

The bottom thermal insulation layer, which has very low thermal conductivity, is typically made of diatomaceous earth (Moler), calcium silicate, or vermiculite based materials [7,14]. The thermal insulation layer limits the heat loss through the bottom of the cell effectively and allows for an overall more energy efficient process. However, while the highly porous structure and low thermal conductivity of these materials make them effective at limiting the heat loss, they are also vulnerable to any bath components or volatiles that may penetrate through the carbon cathode blocks. The insulation layer is therefore protected by a layer of refractory material, which normally is based on aluminosilicates with better resistance towards chemical attack by bath or volatiles species (e.g., Na or $NaAlF_4$) inherent in the production process. Although usually an effective barrier against bath components, the diffusion mechanism of volatile species through the refractories have been identified in spent pot lining by Tschöpe et al. [5]. A first reaction front was found in spent refractories from cell autopsies, where sodium vapor was identified to be the first volatile species to diffuse through the refractory layer. A second reaction front with high fluorine content was also identified, trailing the first reaction front by 1–2 cm. The sodium content in the first reaction layer was found to be as high as 16–18 wt % within the first 3–4 mm. In addition, Tschöpe et al. found that the sodium reaction front had penetrated through 75–100% of the refractory layer in cells 1569–2168 days old, demonstrating the ability of sodium vapor to reach the insulation layer.

In this work, three commercial thermal insulating materials were exposed to sodium vapor by a laboratory test to investigate the impact on the materials chemical and mineralogical durability as well as structural stability. A preliminary study of the chemical stability of the Moler, calcium silicate, and vermiculite based materials has previously been reported [15]. Here, a detailed investigation of mineralogical, chemical and microstructural transformations in the materials is presented, and the findings are compared with reactions predicted by computational thermodynamics and discussed with respect to appropriate ternary phase diagrams in relation to the formation of liquid phases.

## 2. Materials and Methods

The insulation materials investigated were Moler (SUPRA), calcium silicate (SUPER-1100 E), and vermiculite (V-1100 (475)), all products by Skamol A/S. The typical chemical composition of these materials, summarized in Table 1, is quite different, but common for all is the high silica content.

**Table 1.** Typical chemical composition of the insulation materials, from the respective data sheets, given in wt % and loss on ignition (LOI) at 1025 °C.

| Product | $SiO_2$ | $Al_2O_3$ | $Fe_2O_3$ | MgO | CaO | $Na_2O$ | $K_2O$ | $SO_3$ | $TiO_2$ | LOI |
|---------|---------|-----------|-----------|-----|-----|---------|--------|--------|---------|-----|
| **SUPRA** | 77 | 9.0 | 7.0 | 1.3 | 0.8 | 0.4 | 1.6 | 1.0 | 0.7 | 1.0 |
| **SUPER-1100 E** | 47 | 0.3 | 0.3 | 0.6 | 43 | 0.1 | 0.1 | N/A | N/A | 8 |
| **V-1100 (475)** | 46 | 7.0 | 5.5 | 19.0 | 3.5 | 0.2 | 10.0 | N/A | 0.7 | 7.0 |

The laboratory test, named Na vapor test, was inspired by Allaire et al. [16] and conducted at Skamol A/S. Bar-shaped samples were placed inside a steel box with one of their surfaces facing a carbon crucible. 200 g NaF and 100 g Al were used as reactants in the carbon crucible to form cryolite and sodium vapor, by Equation (1)

$$6NaF(s) + Al(l) = Na_3AlF_6 + 3Na(g) \tag{1}$$

The steel box was placed inside an oven, heated to 970 °C, and held at this temperature for 48 h. The Moler material was only heated to 850 °C, due to a limited service temperature of 950 °C. A thermal reference sample was also made for each material by exposing them to the same temperature and length of the Na vapor test but without sodium exposure.

The samples from the laboratory test were visually inspected and documented by optical images. The qualitative phase composition was investigated by powder X-ray diffraction (Bruker AXS D8 Focus) with a LynxEye detector. The step size was 0.014° with a counting time of 0.5 s in the 2θ range of 7° to 70°. The microstructure and chemical composition were investigated by using fracture surfaces as well as polished surfaces of the exposed samples by scanning electron microscopy (SEM) and energy-dispersive X-ray spectroscopy (EDS), using the Zeiss Ultra 55, Limited Edition scanning electron microscope. The polished samples were made by embedding the materials in epoxy and subsequently grinding the surface smooth by diminishing sizes of polycrystalline diamonds down to 1 μm. The EDS measurements were performed on fracture surfaces of the different samples, with three scans at each depth and intervals of 500 μm for the vermiculite and calcium silicate materials, while linescans were performed for the Moler material with measurement intervals of 1 μm across the reaction layer and 10 μm beyond the reaction layer.

The chemical stability of the insulation materials with respect to sodium vapor exposure was investigated by thermodynamic calculations using FactSage (version 7.2) [17]. The Equilibrium module was used in combination with the FactPS and FToxid databases to analyze the phase composition at increasing amounts of sodium vapor. The mass of each oxide was input from the values in Table 1 and normalized to a total mass of 100%. The calculations were made with increments of 0.5 g Na (g), and the temperature was set to correspond with the experimental temperature for each material, under ambient pressure and inert atmosphere. Finally, the Phase Diagram module was used in combination with the FactPS and FTOxid databases to construct isotherm, ternary phase diagrams for Moler ($SiO_2$-$Al_2O_3$-$Na_2O$, T = 850 °C) and calcium silicate ($SiO_2$-$CaO$-$Na_2O$, T = 970 °C). Using the phase diagrams, the chemical composition of Moler and calcium silicate were simplified to consist of only $SiO_2$ and $Al_2O_2$, and $SiO_2$ and $CaO$, respectively.

## 3. Results

### 3.1. Laboratory Tests

### 3.1.1. Moler

A cross section of the exposed Moler material is shown in Figure 1, next to the pristine material for comparison. A thin, glassy reaction layer (Rx-layer) formed on the surface closest to the carbon crucible. Below this Rx-layer the brick changed color from the pristine state, but no further macroscopic changes were observed.

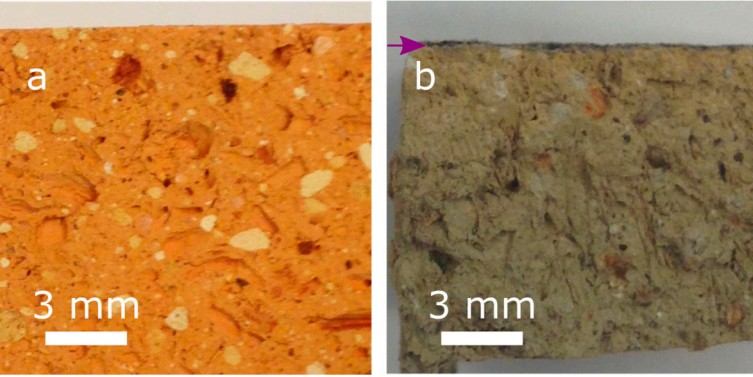

**Figure 1.** Optical image of (**a**) the pristine Moler brick and (**b**) the Moler brick after Na-exposure with a thin, glassy reaction layer at the top surface (purple arrow).

XRD diffractograms of the Moler material are shown in Figure 2. The Moler material has been calcined during manufacturing, and no mineralogical changes were observed between the pristine state and the thermal reference. Quartz ($SiO_2$) and iron oxide ($Fe_2O_3$) are the main crystalline phases

in addition to a significant amount of amorphous or nano-crystalline phase evidenced by the broad background in the diffractograms. Albite ($NaAlSi_3O_8$) was identified in the Rx-layer, while quartz and cristobalite were found beyond the surface Rx-layer. The iron oxide found in the pristine and thermal reference sample was not observed either in the Rx-layer or beyond.

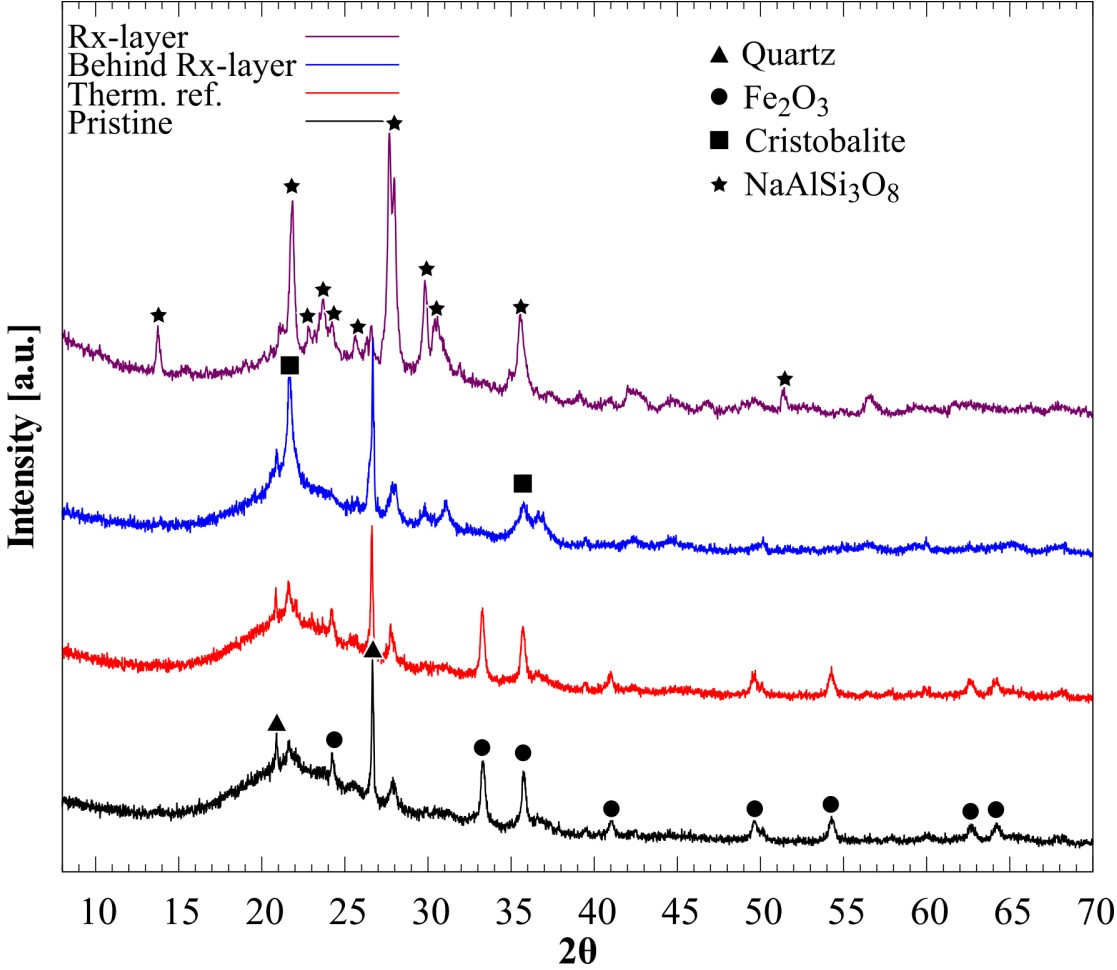

**Figure 2.** XRD diffractograms of the Moler material. From top to bottom: Rx-layer, behind Rx-layer, thermal reference sample, and pristine sample. The most important reflections of the crystalline phases are marked.

An overview of the microstructure of the sodium exposed Moler material is shown in Figure 3a, while higher resolution images at given depths into the interior of the brick are shown in Figure 3b–e. The backscattering SEM images of the outer reaction layer is heterogeneous corresponding to a glass ceramic with elongated crystals with a needle or plate-like shape. EDS analysis demonstrated that certain areas possess higher Na content than others.

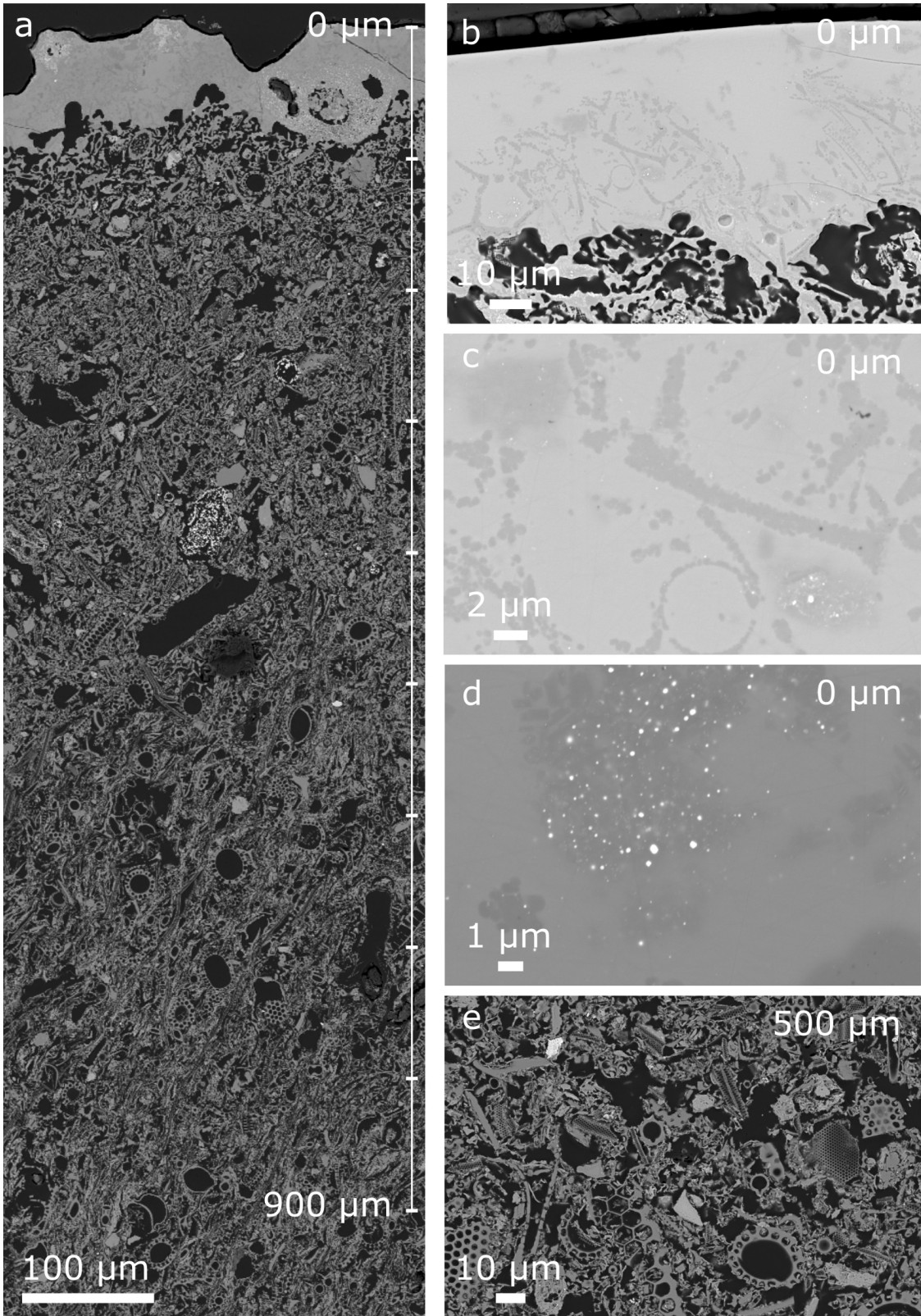

**Figure 3.** (**a**) An overview image of Na-exposed Moler brick with the exposed surface at upper part of the image; (**b**) glassy Rx-layer; (**c**) high resolution of glassy Rx-layer; (**d**) metallic Fe in bright spots; (**e**) typical microstructure of unreacted material.

The decreasing Na content from the surface to the interior of the brick is shown in Figure 4, showing a typical EDS linescan across the Rx-layer (insert) with Na content as a function of depth into the brick. The inhomogeneous Na content through the reaction layer due to the complex microstructure is also reflected in Figure 4. The thickness of the Rx-layer varied at different positions in the brick with a maximum thickness of 200 µm. In addition, small bright particles, as seen in Figure 3d, were confirmed by EDS to correspond to precipitated metallic iron in the Rx-layer.

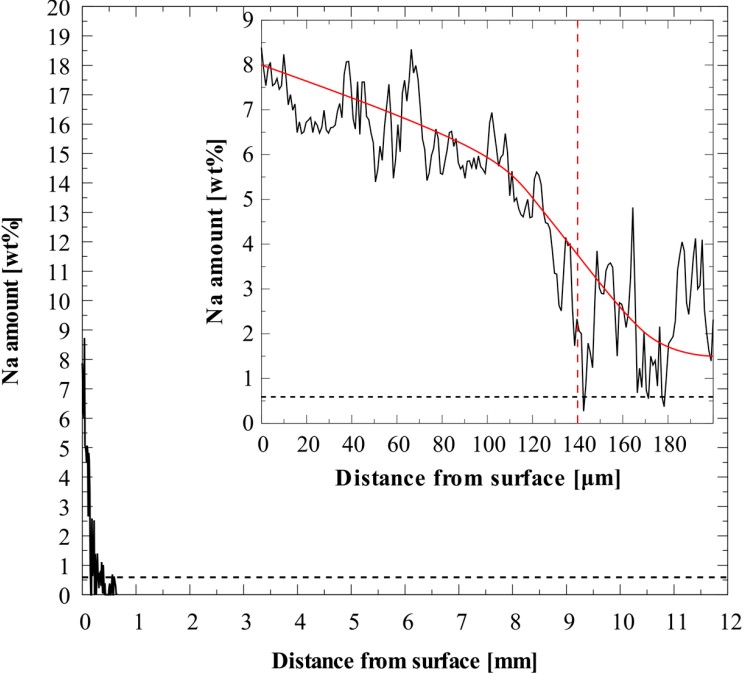

**Figure 4.** EDS linescan of a polished Moler sample from the exposed surface and 640 µm into the brick. Horizontal black dotted line gives the Na content in the pristine material. Inset: A typical EDS linescan of fracture surface from outer surface towards the interior of the Moler material. Red, vertical dotted line marks end of glassy layer.

### 3.1.2. Calcium Silicate

The calcium silicate material was deformed by creep during the Na vapor test, forming a reaction zone of approximately 1 cm. This Rx-zone can be seen divided into an outer, mid, and inner Rx-layer in Figure 5. The pristine material is included in the figure for comparison.

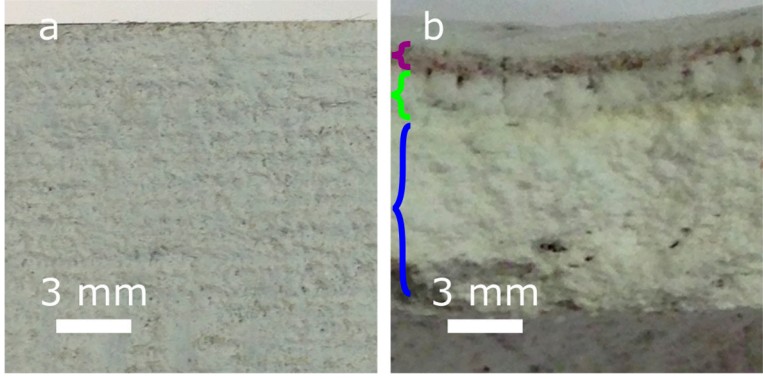

**Figure 5.** Optical images of (**a**) pristine calcium silicate brick, and (**b**) calcium silicate brick after Na-exposure. Upper surface in the image is the exposure surface. The reaction zone is divided into an outer (purple bracket), mid (green bracket), and inner (blue bracket) reaction layer.

The XRD diffractograms of the calcium silicate material are shown in Figure 6. The pristine material consisted of xonotlite (Ca6Si$_6$O$_{17}$(OH)$_2$), while the thermal reference was converted to wollastonite (CaSiO$_3$) by thermal treatment. Na$_2$Ca$_2$Si$_2$O$_7$ was identified in the outer Rx-layer, while Na$_2$Ca$_2$Si$_3$O$_9$ was found in addition in the middle part of the Rx-layer. Wollastonite and Na$_2$Ca$_2$Si$_3$O$_9$ were the main crystalline phases in the inner Rx-layer, while only wollastonite was found in the interior of the brick beyond the Rx-layer.

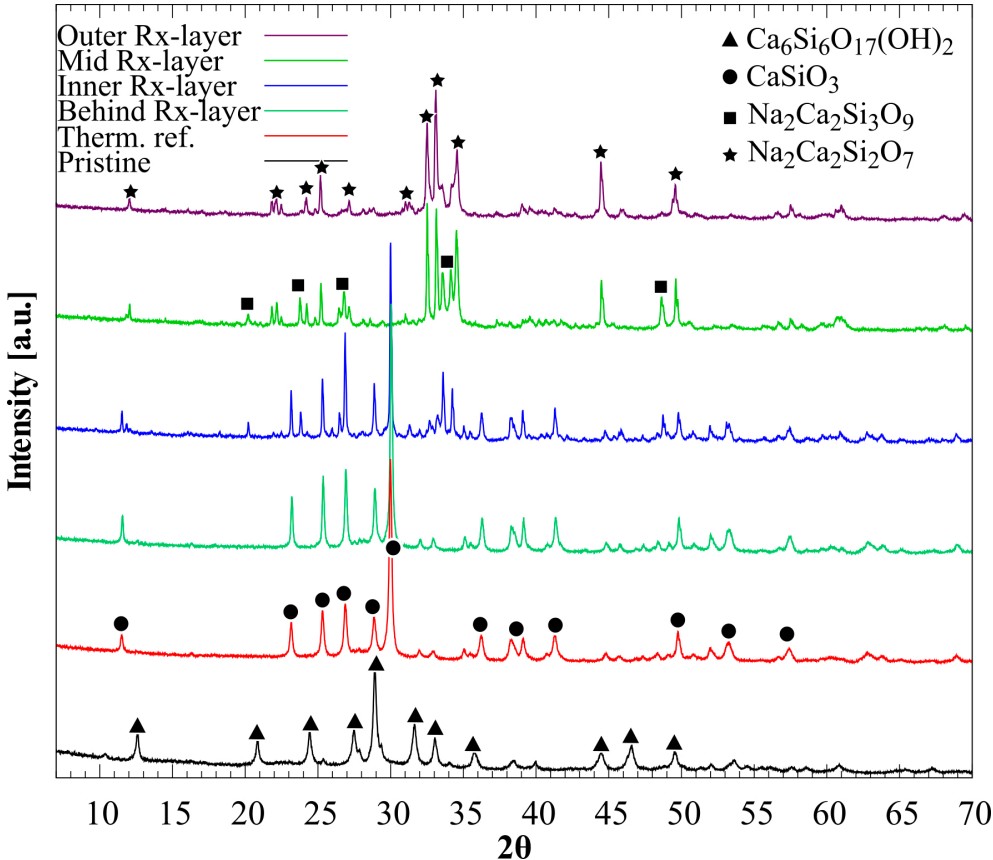

**Figure 6.** XRD diffractograms of the calcium silicate material. From top to bottom: Outer Rx-layer, mid Rx-layer, inner Rx-layer, behind Rx-layer, thermal reference sample, and pristine sample. The most important reflections due to crystalline phases are marked.

An overview of the microstructure of the sodium exposed calcium silicate material is shown in Figure 7a. High-resolution images of the fracture surface at given depths into the material are shown in Figure 7b–e. The needle-like microstructure characteristic of pristine calcium silicate is observed below the Rx-zone, 1 cm into the brick, see Figure 7e. The needles are only a few hundred nm in diameter. A significant coarsening of the microstructure is observed closer to the outer surface, being closest to the sodium vapor source. The diameter of the needles has increased as much as an order of magnitude from e to c, and at the outer Rx-layer in b the microstructure is much coarser and no longer resembling the needle-like network.

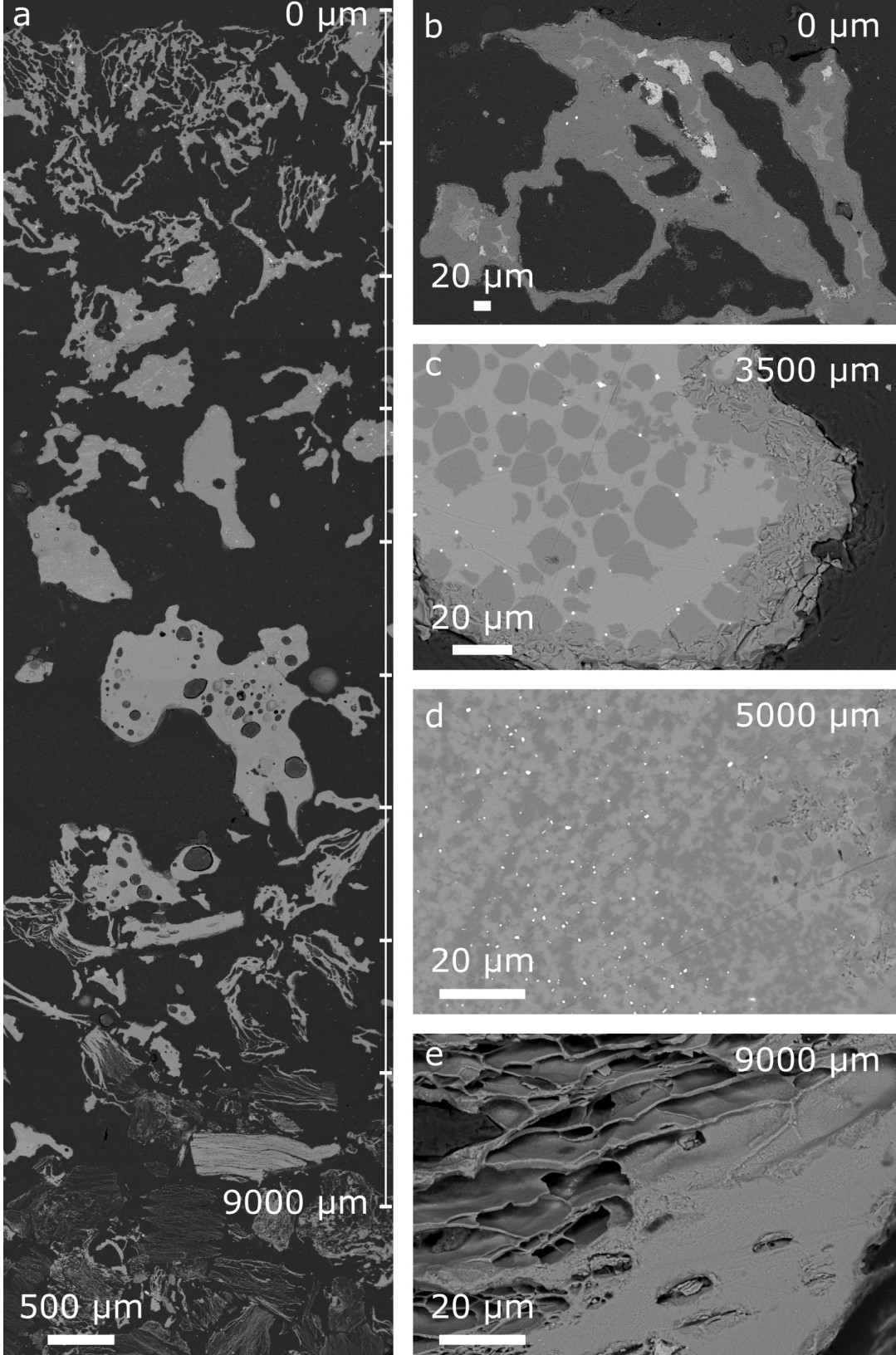

**Figure 7.** (**a**) An overview image of a Na-exposed calcium silicate sample with the exposed surface at upper part of the image; (**b**–**d**) microstructure of fracture surfaces at increasing depths into the reaction zone. Typical microstructure of unreacted material shown in (**e**).

The Na content measured by EDS in the calcium silicate material as a function of the distance from the exposed surface is shown in Figure 8. The Na content is generally high throughout the Rx-zone, and the standard deviations of the measurements are significant due to the complex microstructure and porosity. One centimeter (10,000 μm) into the sample, the Na content drops to levels indistinguishable from the pristine material.

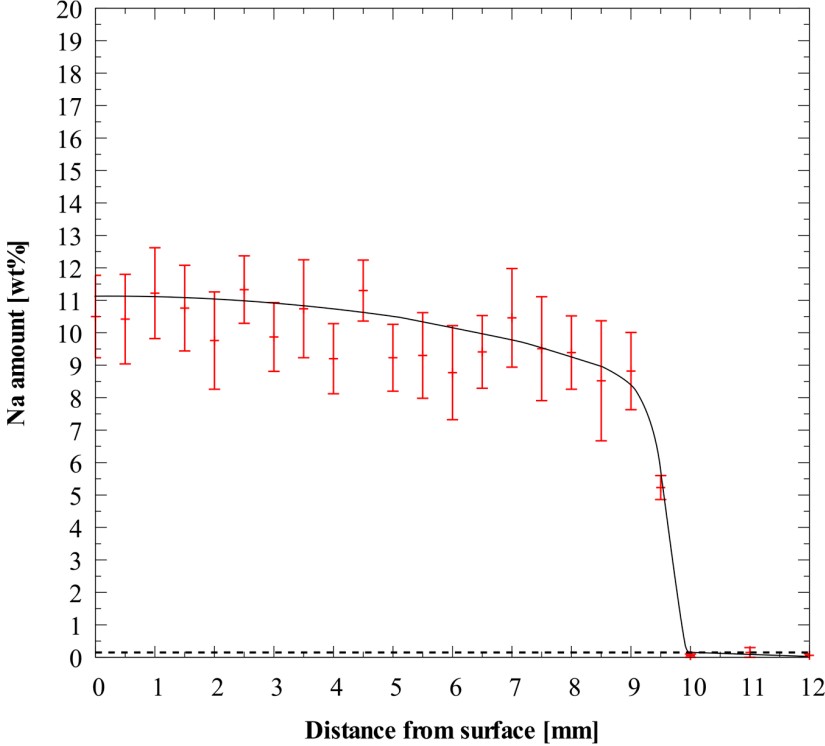

**Figure 8.** Na content across the reaction zone of the calcium silicate material. Black dotted line gives the Na content in pristine material.

### 3.1.3. Vermiculite

The vermiculite material formed a total reaction zone of an approximate depth of 9 mm, which is divided into an outer and inner Rx-layer, as shown in the optical image in Figure 9. Behind the Rx-zone, the color of the sodium exposed sample changed from the pristine state, but with no further macroscopic changes observed.

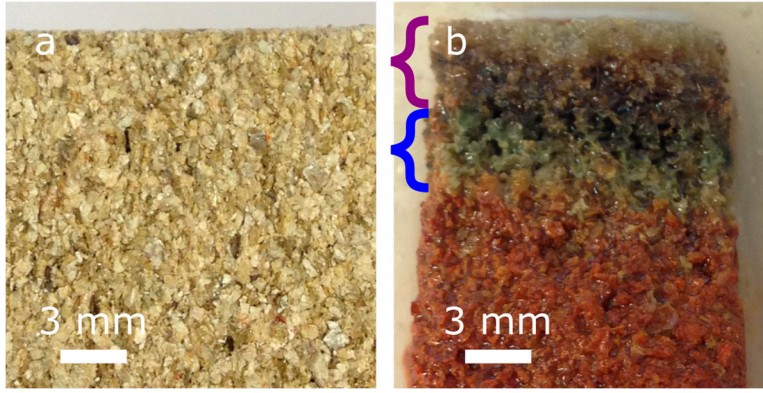

**Figure 9.** Optical images of (**a**) pristine vermiculite brick and (**b**) vermiculite brick after Na-exposure, top surface in the image is exposure surface. The reaction zone has been divided into an outer (purple bracket), and inner (blue bracket) reaction layer.

The XRD diffractograms of the vermiculite material are shown in Figure 10. The pristine material consists of the crystalline phase phlogopite, while forsterite ($Mg_2SiO_4$) and leucite ($KAlSi_2O_6$) are found in addition in the thermal reference. Forsterite and leucite are also the major phases identified behind the Rx-zone in the sodium exposed sample. $(Na_2O)0.33 \cdot NaAlSiO_4$ was found in the outer Rx-layer, while forsterite and kaliophilite ($KAlSiO_4$) were identified in the inner Rx-layer.

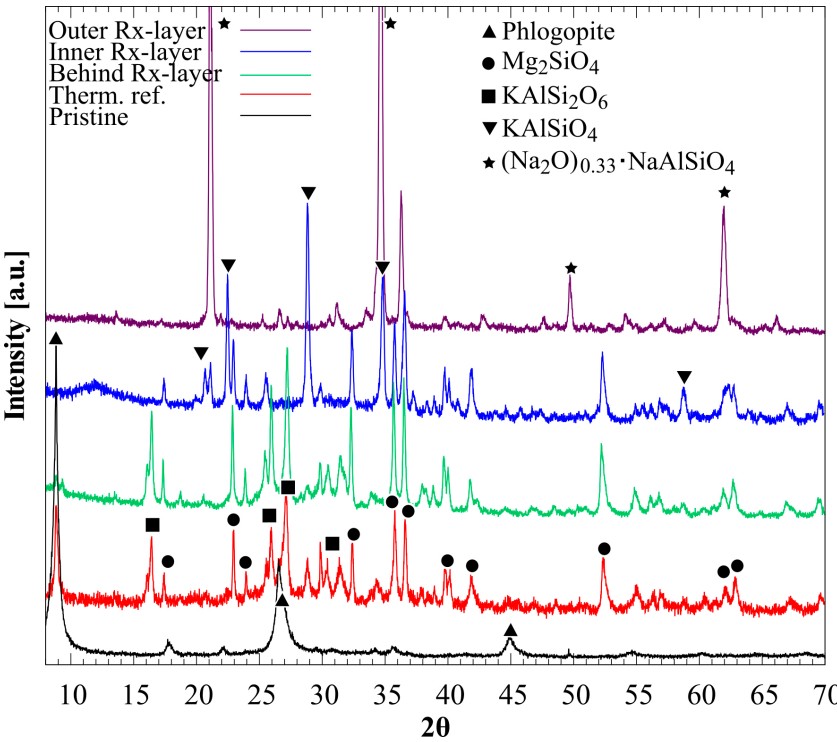

**Figure 10.** XRD diffractograms of the vermiculite material. From top to bottom: Outer Rx-layer, inner Rx-layer, behind Rx-layer, thermal reference sample, and pristine sample. The most important reflections are marked.

The microstructure of the sodium exposed vermiculite material is shown in Figure 11, with an overview in (a), and higher resolution images at certain depths into the sample in (b–e). The characteristic parallel sheet microstructure of vermiculite grains can be observed in the interior behind the Rx-zone in Figure 11e. This sheet microstructure was no longer present in the Rx-zone. The severe

coarsening of the microstructure in Figure 11a, especially in the area 2–7 mm from the surface, suggests the formation of a liquid phase during the sodium exposure.

The Na content in the sodium exposed vermiculite material, as measured by EDS, is shown in Figure 12. The Na content is generally high throughout the Rx-zone and drops to levels indistinguishable from the pristine material from 9.5–10 mm inwards. The scatter in the data are large, reflecting the heterogeneous nature of the reaction zone as can be seen in Figure 11c,d.

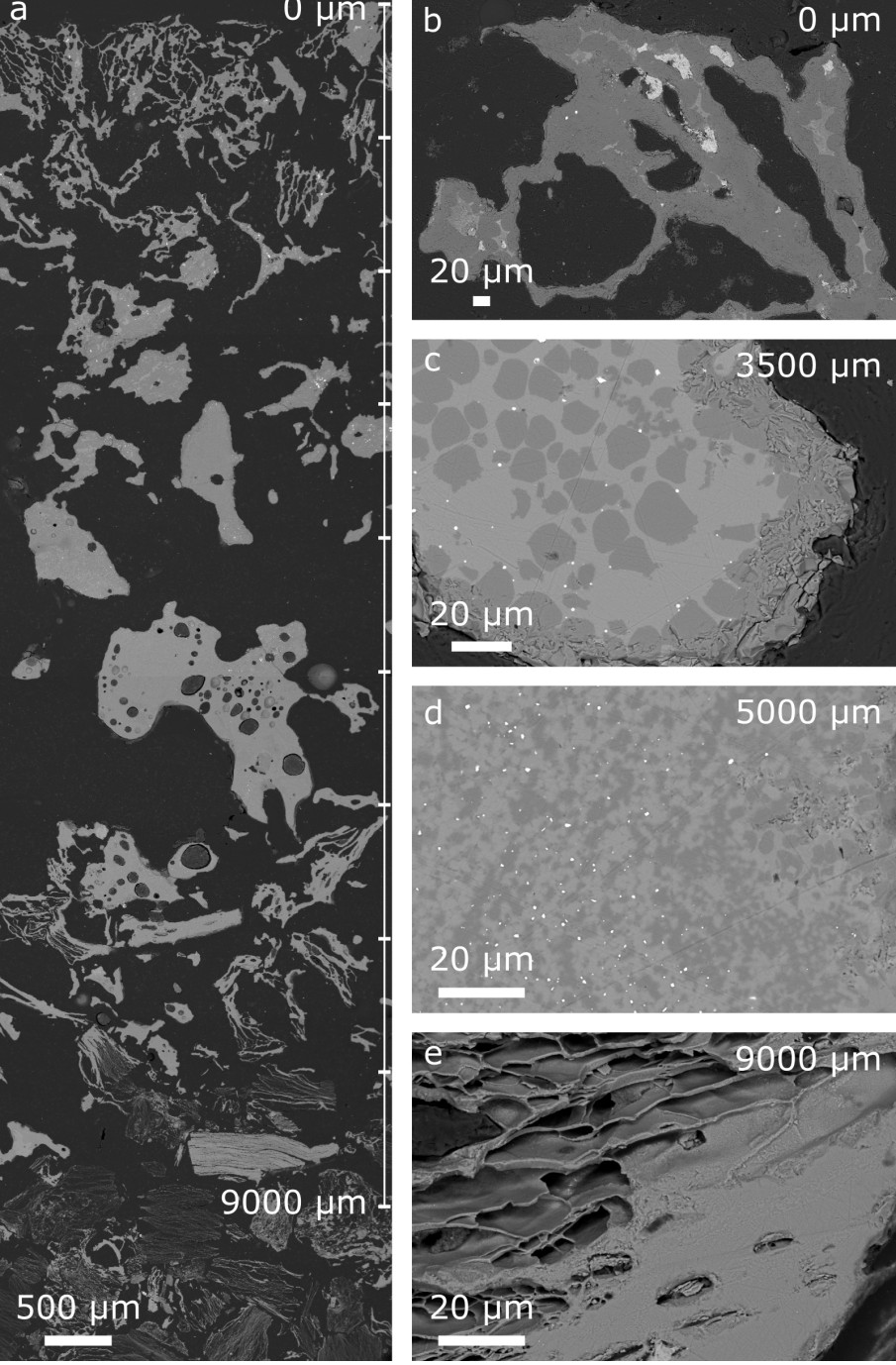

**Figure 11.** (**a**) An overview SEM image of a Na-exposed vermiculite sample with the exposed surface at upper part of the image. (**b**–**e**) higher resolution images showing the microstructure at different depths into the reaction zone.

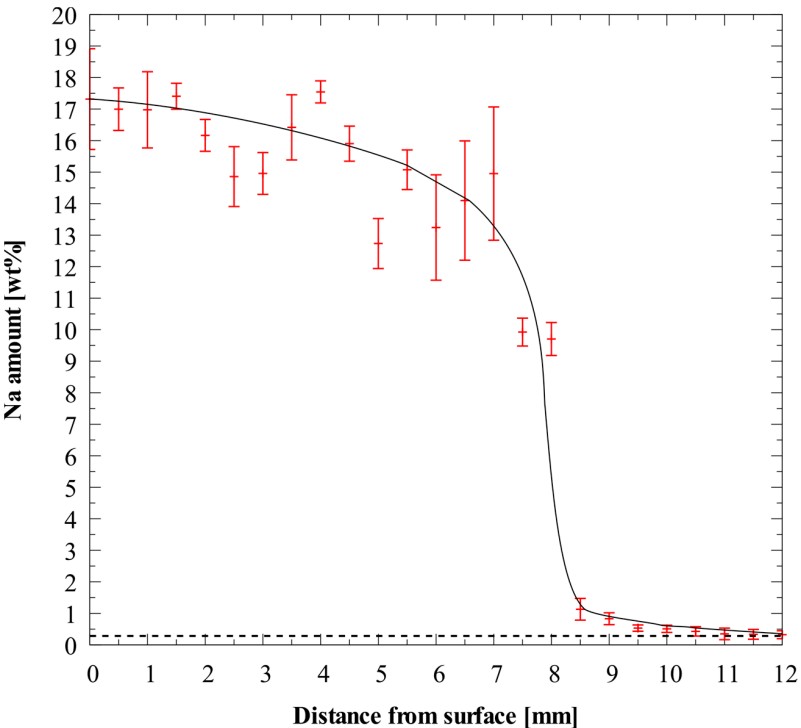

**Figure 12.** Na content across the reaction zone of the vermiculite material. Na content in the pristine material given by the horizontal dotted line.

## 3.2. Thermodynamic Calculations

### 3.2.1. Moler

The calculated stable phases present in the case of Moler material with increasing amounts of sodium are shown in Figure 13. The most important stable phases present before Na exposure are $SiO_2$, $KAlSi_3O_8$, $Mg_2Al_4Si_5O_{18}$, and $Fe_2O_3$. An increasing amount of albite ($NaAlSi_3O_8$) is formed up to 3 wt % Na, followed by the formation of two different slag phases. The three most abundant components in the two slag phases are given for Na-content 1 and 2 (Figure 13) in Table 2. The pristine iron oxide in Moler is reduced to metallic iron and further to $Fe_3Si$ with increasing Na content.

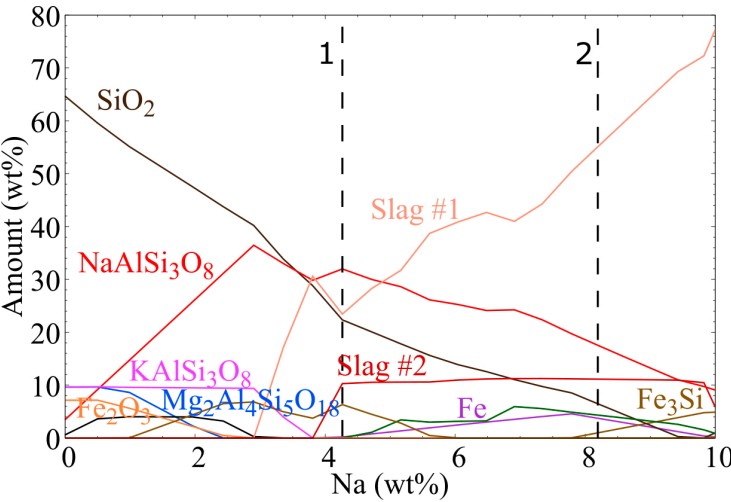

**Figure 13.** Phase composition (excluding minor phases below 4 wt %) of Moler with increasing amounts of Na. Formation of slag #1 at ~3 wt % Na, and slag #2 at ~4 wt % Na.

**Table 2.** The content of the major components in the two slag phases formed in the Moler material at a Na-content according to 1 and 2 in Figure 13.

| | Point 1 | Point 2 |
|---|---|---|
| **Slag #1** | $SiO_2 = 73.8$<br>$KAlO_2 = 11.0$<br>$NaAlO_2 = 8.0$ | $SiO_2 = 72.5$<br>$NaAlO_2 = 11.1$<br>$Na_2O = 9.7$ |
| **Slag #2** | $SiO_2 = 83.8$<br>$Na_2O = 9.0$<br>$TiO_2 = 6.4$ | $SiO_2 = 83.4$<br>$Na_2O = 9.5$<br>$TiO_2 = 5.8$ |

Slag #1 is a silica-rich slag that has comparable amounts of K and Na at point 1, but with much higher Na content at point 2. Slag #2 is also silica-rich and contains considerable amounts of $Na_2O$ and $TiO_2$, but changes little in composition from point 1 to point 2.

### 3.2.2. Calcium Silicate

The equilibrium phase content in cases of calcium silicate with increasing amounts of sodium are shown in Figure 14. Wollastonite ($CaSiO_3$) is the major phase present before Na is introduced. Wollastonite is consumed in the reaction with Na, where $Ca_3Si_2O_7$ and $Na_4CaSi_3O_9$ are formed, and lastly, there is the formation of $Na_2CaSiO_4$. Only minor slag formation is predicted. Metallic silicon is formed by the reducing conditions.

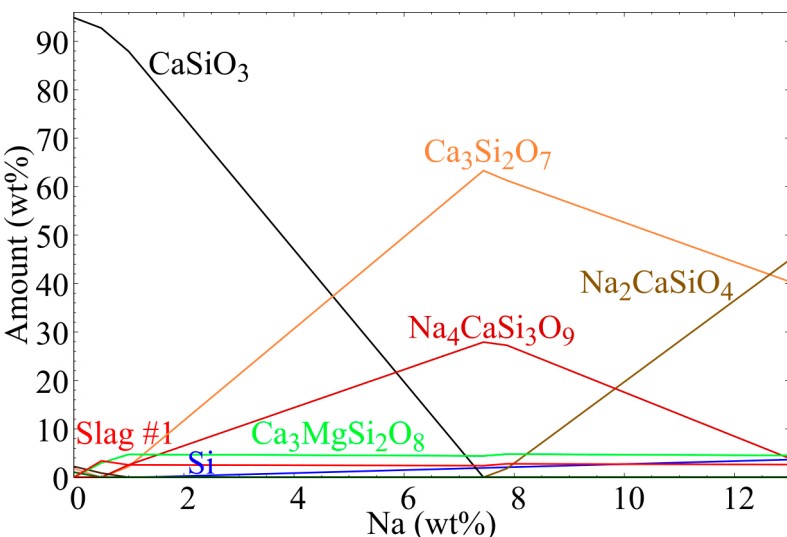

**Figure 14.** Phase composition of calcium silicate with increasing amounts of Na.

### 3.2.3. Vermiculite

The equilibrium phase composition in vermiculite with increasing amounts of sodium are shown in Figure 15. The major initial phases are predicted to be forsterite ($Mg_2SiO_4$), leucite ($KAlSi_2O_6$), $CaMgSi_2O_6$, $Fe_2O_3$, and a slag. A second slag is formed with small amounts of Na (<2 wt %), but with increasing amounts of Na, the formation of slag #1 is dominant at the expense of the initial phases. The abundance of the three most important species in the slag phases are given for point 1 and 2 in Table 3. The initial iron oxide is reduced to metallic iron and other iron-containing phases due to the reducing conditions with increasing Na content.

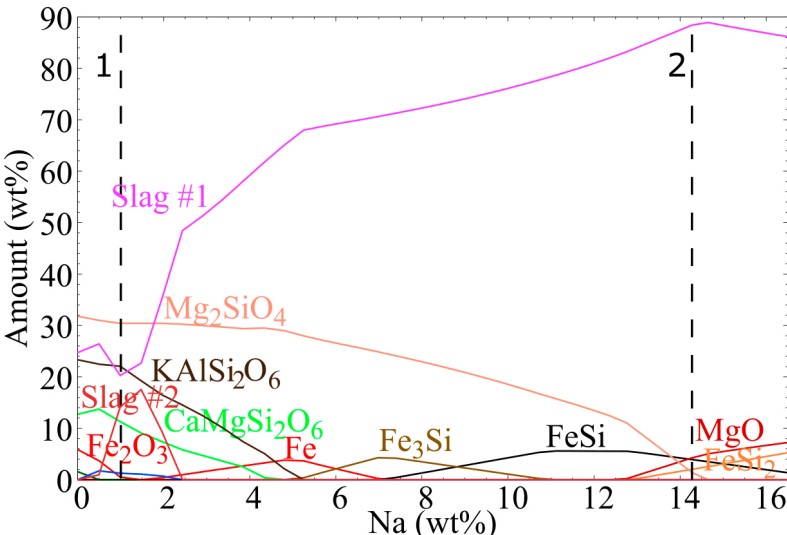

**Figure 15.** Phase composition of vermiculite with increasing amounts of Na. Major slag formation at low Na content. Reduction of iron oxide to other iron species.

**Table 3.** wt % of the three major slag species at point 1 and 2 from Figure 15, vermiculite material.

|          | Point 1              | Point 2             |
| -------- | -------------------- | ------------------- |
| **Slag #1** | $SiO_2$ = 59.7<br>$K_2O$ = 17.6<br>$KAlO_2$ = 15.8 | $SiO_2$ = 41.6<br>$Na_2O$ = 19.4<br>$MgO$ = 14.5 |
| **Slag #2** | $SiO_2$ = 42.6<br>$FeO$ = 14.7<br>$NaFeO_2$ = 10.9 |  |

Slag #1 is silica-rich, with predominantly potassium at point 1, while sodium and magnesium are more present at point 2. Slag #2 is also silica-rich, but with significant amounts of iron and some sodium.

## 4. Discussion

Even though no external load was applied to the materials during testing, Na exposure had distinctly different effects on the macrostructure of the thermal insulating materials. The Moler and vermiculite bricks had only minor macroscopic structural changes, while the calcium silicate deformed by creep under its own weight. The loss of structural integrity of the insulation layer will have severe effects in the electrolysis cell as it supports the layers above it. Paulsen et al. [9] reported two creep steps for pristine calcium silicate under load, at 700 °C and 900 °C, both below the temperature used in this work. The maximum temperature in the insulation layer in electrolysis cells under operation is expected to be well below 900 °C [9,10]. However, the severe deformation observed in this work cannot be attributed to temperature alone, as the temperature reference samples were not deformed in the same manner, meaning Na degradation is the most important factor. The combined effect of Na exposure and temperature should, therefore, be investigated more thoroughly, especially for calcium silicate as the creep rate is likely to decrease with decreasing temperature. In a different laboratory test by the authors, deformation by creep was not observed for calcium silicate at a temperature of 800 °C, demonstrating that the onset of creep due to Na exposure is in the temperature interval 800–970 °C [18]. At the conditions used in this study, Moler and vermiculite are clearly structurally more stable than calcium silicate when exposed to Na vapor.

The penetration depth of sodium varies in the tested materials from a few hundred μm in Moler to about 1 cm for calcium silicate. It is important to note that the sodium activity was lower in the test procedure of Moler due to the lower temperature. Nevertheless, the sodium was mostly concentrated in the glass-ceram Rx-layer, and a gradient in the Na content was observed (Figure 4), showing that the glassy layer is reducing further penetration of Na as a protective layer of glass-ceram is built up. However, it is important to note that due to the low initial density of the bricks, a considerable thickness of the brick will be consumed by the reaction, which will severely reduce the thermal insulating capability. The Na content is high throughout the Rx-zones of both vermiculite (Figure 12) and calcium silicate (Figure 8), demonstrating that there is no barrier formed, and further Na penetration in these two materials will not be hindered at the given conditions.

The mineralogy of the materials changes significantly with Na exposure. In the case of vermiculite and calcium silicate, the changes are also due to the thermal treatment as these products are not fired to high temperatures during manufacturing. Moler, a fired product, has no observable change in mineralogy when fired to 850 °C. When fired to 970 °C, forsterite and leucite are found in the vermiculite material, and a complete mineralogical transformation from xonotlite to wollastonite was observed for calcium silicate, demonstrating that these materials are not fired to their chemical equilibrium during manufacturing. This is in agreement with similar findings of Paulsen et al. [9]. Knowledge of the mineralogy of the insulation bricks at operating temperature is important as it may have different materials properties from the ones reported in the non-fired state.

When exposed to Na vapor, albite was found in the glassy Rx-layer of Moler, which is in agreement with the thermodynamic calculations (Figure 13), which also predicted the formation of albite. At higher concentration of Na, albite is consumed, resulting in the formation of two slag phases. Albite has a strong tendency to form glass and is, therefore, not usually found in its crystalline state [19]. Furthermore, iron oxide is reduced to metallic iron under the reducing conditions of the test, also in good agreement with the thermodynamic calculations. Metallic silicon has been observed to form in the refractory material above the insulation layer both in laboratory testing and samples from spent potlining [5,20,21]. Assuming a simplified chemical composition of the Moler brick, where only $SiO_2$ and $Al_2O_3$ are considered, the isothermal ternary phase diagram of $SiO_2$-$Al_2O_3$-$Na_2O$, shown in Figure 16, can be used to evaluate the reaction path during Na exposure. The initial composition is given by the red dot at 89.5 wt % $SiO_2$ and 10.5 wt % $Al_2O_3$. With an increasing amount of sodium, the composition pathway follows the dotted line. The coexisting phases in the marked regions are given in Table 4. Slag formation is clearly unavoidable at the given conditions and initiates at only a few wt % of $Na_2O$. Albite is predicted to form at the initial stage, and albite is known to not crystallize easily. Formation of a viscous liquid or slag [22] in Moler exposed to Na is therefore likely even at a low amount of Na.

**Table 4.** Coexisting phases in the given regions of the ternary phase diagram in Figure 16.

| Region Number | Coexisting Phases |
|:---:|:---:|
| a | $SiO_2$<br>$NaAlSi_3O_8$<br>$Al_2O_3$ |
| b | $SiO_2$<br>Slag (liq)<br>$NaAlSi_3O_8$ |
| c | $SiO_2$<br>Slag (liq) |
| d | Slag (liq) |
| e | Slag (liq)<br>$NaAlSi_3O_8$ |

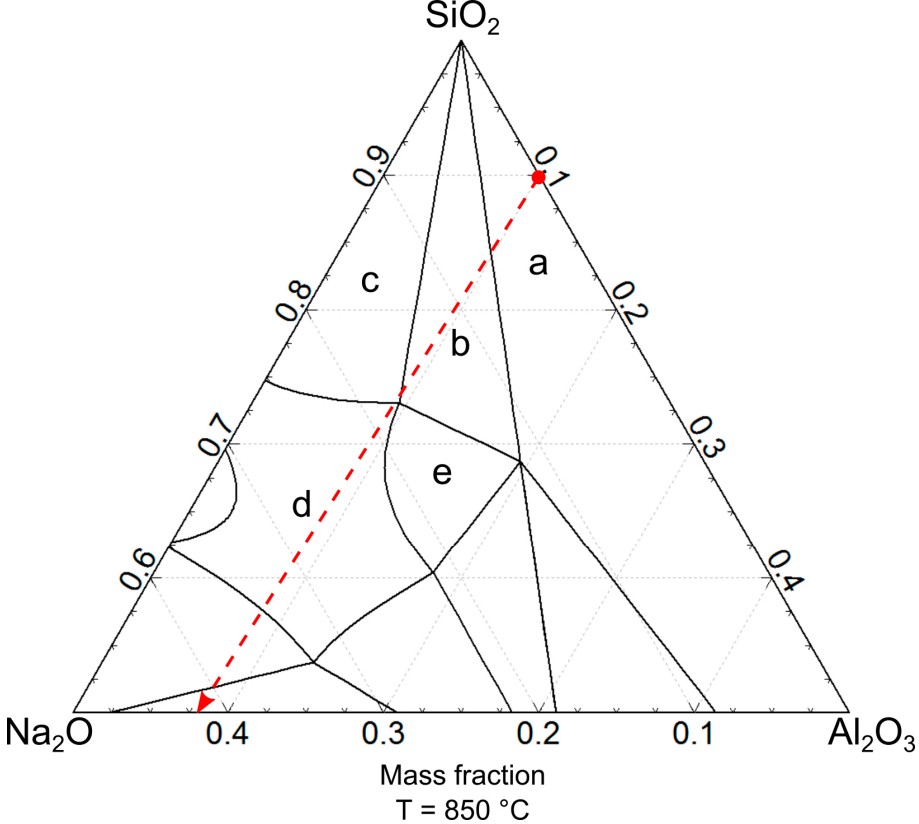

**Figure 16.** 850 °C isotherm in the SiO$_2$-Al$_2$O$_3$-Na$_2$O system. The red dot marks the initial, simplified starting point of the Moler material, while the dotted line projects increasing amounts of sodium. The coexisting phases for each marked region are given in Table 4.

Na$_2$C$_{a2}$Si$_2$O$_7$ and Na$_2$Ca$_2$Si$_3$O$_9$ were the two main Na-containing crystalline phases observed in the Rx-zone of calcium silicate. However, thermodynamic calculations (Figure 14) predict the formation of the two phases Na$_4$CaSi$_3$O$_9$ and Na$_2$CaSiO$_4$. The deviation between the experiments and thermodynamics demonstrates that the kinetics of the reaction with Na are also important since the predicted equilibrium phase composition was not observed. Na$_2$CaSiO$_4$ has been observed in a different laboratory test at 800 °C, which is lower than the current temperature of the test [18]. SiO$_2$ and CaO are the two main principle oxides in calcium silicate (Table 1). The isothermal SiO$_2$-CaO-Na$_2$O ternary phase diagram at 970 °C is given in Figure 17, showing the initial chemical composition by the red dot. The reaction path with increasing Na content follows the dotted line. The coexisting phases in the marked regions are given in Table 5. In this case, there is no slag formation, and the possible Na-containing phases are Na$_4$CaSi$_3$O$_9$ and Na$_2$CaSiO$_4$, as predicted by the equilibrium calculations (Figure 14), but not observed by the current experiment (Figure 6).

Beyond the observable Rx-layer of Na-exposed Moler, cristobalite is identified by XRD (Figure 2). The diffraction lines had previously been thought to identify the phase Mg$_2$Al$_4$Si$_5$O$_{18}$ [15]. The conversion of quartz to cristobalite has been reported at temperatures close to the testing temperature in this work, in the presence of an alkali source such as sodium [23].

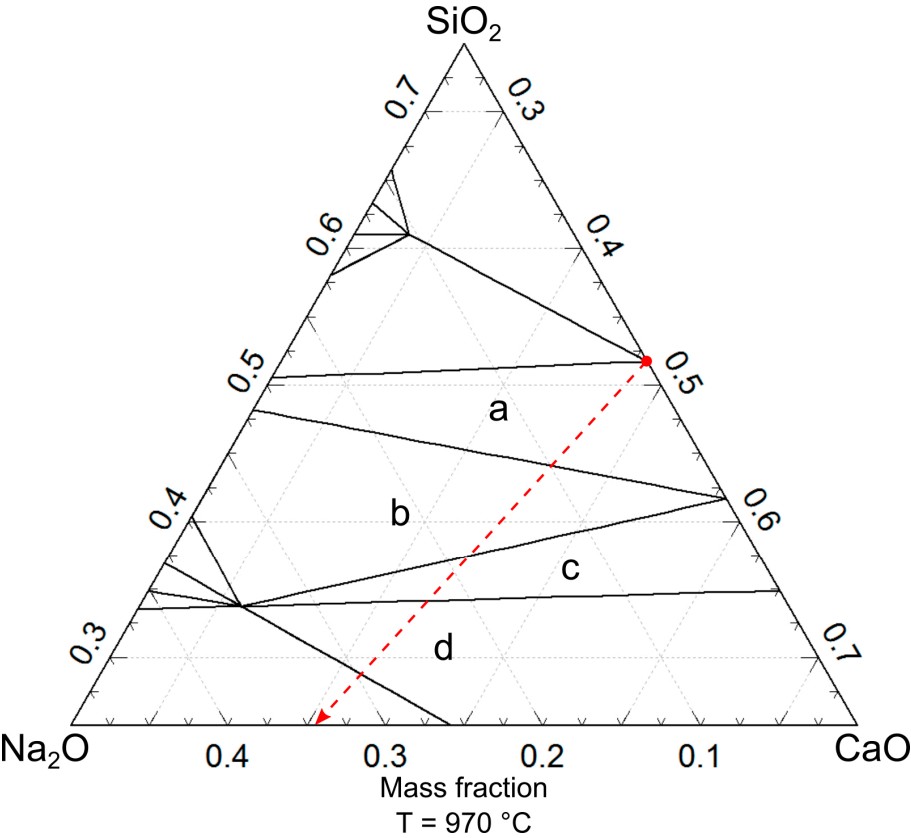

**Figure 17.** 970 °C isotherm in the SiO$_2$-CaO-Na$_2$O system. The red dot marks the initial, simplified starting point of the calcium silicate material, while the dotted line projects increasing amounts of sodium. The coexisting phases for each marked region are given in Table 5.

**Table 5.** Coexisting phases in the given regions of the ternary phase diagram in Figure 17.

| Region Number | Coexisting Phases |
|---|---|
| a | CaSiO$_3$<br>Na$_4$CaSi$_3$O$_9$<br>Ca$_3$Si$_2$O$_7$ |
| b | Na$_4$CaSi$_3$O$_9$<br>Na$_2$CaSiO$_4$<br>Ca$_3$Si$_2$O$_7$ |
| c | Ca$_3$Si$_2$O$_7$<br>Na$_2$CaSiO$_4$<br>Ca$_2$SiO$_4$ |
| d | Ca$_2$SiO$_4$<br>Na$_2$CaSiO$_4$<br>CaO |

Phlogopite, forsterite, and leucite are present in the thermal reference sample of vermiculite, in fair agreement with the thermodynamic equilibrium calculations (Figure 15). However, the calculations predict significant slag formation with increasing Na content, even before exposure to Na. The severe coarsening of the microstructure (Figure 11) does indicate the presence of a viscous phase during the test as this change in microstructure is unlikely to occur by solid state diffusion alone. Nepheline ((Na$_2$O)0.33 · NaAlSiO$_4$) and kaliophilite (KAlSiO$_4$) are identified in the outer and inner Rx-layers, respectively. These phases are not predicted by the present calculations, however, they were predicted

in calculations not including the FTOxid database [17]. No Na-containing crystalline phase was found in the inner Rx-layer, which indicates that the sodium in this layer must be present in an amorphous phase, supporting the prediction of slag formation. Phlogopite and leucite reacted first with Na before forsterite when the material is exposed to sodium, showing forsterite to be more chemically stable under the current conditions. This is also predicted by the thermodynamic calculations. Similarly to Moler, metallic iron (bright spots in Figure 11d is found in the Rx-zone of vermiculite as iron oxide is reduced.

## 5. Conclusions

Sodium is the first volatile specie capable of reaching the insulating layer in a normal operating Hall-Héroult electrolysis cell. Therefore, a laboratory scale test was set up to expose three commercial thermal insulating materials (Moler, calcium silicate, and vermiculite) to sodium vapor. Calcium silicate deformed by creep under no external load, while Moler and vermiculite were much more structurally stable. The microstructure of the exposed materials changed significantly in the reacted areas, with the formation of a heterogeneous glass-ceram in Moler, microstructure coarsening and an apparent liquid phase in vermiculite, and a dramatic coarsening of the needle-like microstructure in calcium silicate. The sodium penetration depth varied from ~200 μm in Moler to 9 and 10 mm in vermiculite and calcium silicate, respectively. As the experimental temperature was lower for Moler than for calcium silicate and vermiculite, it is not possible to do a direct comparison. However, the sodium gradient in the Moler reaction layer suggests that further sodium penetration may be reduced by the viscous glass-ceram present. Finally, thermodynamic equilibrium calculations were performed on the three different materials, largely predicting the formation of the same phases observed experimentally. The exception was calcium silicate, demonstrating that reaction kinetics are also important in the reaction with sodium vapor.

**Author Contributions:** R.L. conducted all the analysis, thermodynamic calculations and wrote the paper; S.N.B. conducted the tests and with conceptualization; J.M. supervised and conceptualization the study; A.P.R. contributed with conceptualization, project administration and supervision of the study. T.G. contributed with conceptualization, supervision and original draft preparation.

**Funding:** This research was funded by the Norwegian Research Council, grant number 236665, and the partners Hydro Aluminium, Alcoa, Elkem Carbon, and Skamol through the project "Reactivity of Carbon and Refractory Materials used in metal production technology" (CaRMa).

**Acknowledgments:** Skamol is acknowledged for the support to conduct the test and providing materials for the study.

**Conflicts of Interest:** The authors declare no conflict of interest.

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
