# Peer review of "Chemical Durability of Thermal Insulating Materials in Hall-Héroult Electrolysis Cells"

_ceramics, doi:10.3390/ceramics2030034_

Round 1

Reviewer 1 Report

The basic goal of the article was to conduct research in terms of lowering the Hall-Heroult process operating costs by reducing heat losses through an insulating refractory lining layer at the bottom of it. The insulation layer is protected by a layer of refractory material, which normally is based on aluminosilicates with good resistance towards chemical attack by bath or volatiles species (e.g. Na or NaAlF4) inherent in the production process. The thermal insulation layer is critical for the overall thermal stability of the cell and is vulnerable to volatile species, such as sodium vapour, that may penetrate through the carbon cathode and refractory layer.

The paper presented an investigation of the chemical degradation of typical thermal insulating materials (Moler - diatomaceous earth, calcium silicate, and vermiculite)  by exposure to sodium vapour in a laboratory test to investigate the impact on the materials chemical and mineralogical durability, as well as structural stability. Sodium is the volatile species capable of reaching the insulating layer in a normal operating Hall-Héroult electrolysis cell.

The microstructure and chemical composition were investigated using by optical images, powder X-ray diffraction and fracture surfaces as well as polished surfaces of the exposed samples by scanning electron microscopy (SEM) and energy-dispersive X-ray spectroscopy (EDS).

The obtained results were compared with reactions predicted by computational thermodynamics and discussed with respect to appropriate ternary phase diagrams in relation to the formation of liquid phases.

This allowed the authors to show differences in the behaviour of the materials studied. They proved that the chemical and phase composition, as well as the microstructure of the starting materials, determined their corrosion mechanism under the influence of volatile sodium. They stated that the Moler and vermiculite bricks had only minor macroscopic structural changes, while the calcium silicate deformed by creep under its own weight.

The authors in the discussion described detailed of the mechanism of the reaction of materials and summarize the basic differences in the course of their corrosion.

In my opinion, in summary, there are no practical conclusions as to the suitability of the materials tested for a very responsible insulation lining in the Hall-Heroult installation. The loss of structural integrity of the insulation layer has severe effects in the electrolysis cell as it supports the layers above it. This is a very important technological information for manufacturers and users of refractories.

The article is interesting from a scientific point of view and is suitable for printing after taking into account the above remark and a small text correction, e.g.:

- taking into account the same letter designations - Figure 16 and Table 4 (Region number).

Author Response

We thank the reviewer for the kind evaluation. The letter designations in Figure 16 and table 4 are identical in the revised manuscript. 

Reviewer 2 Report

The following changes should be implemented before the work may be considered for publication:

Please check language/formatting as there are minor issues all through - The Production (L27), Materials The insulation materials (L70), 8 to 8.0 in Table 1, ot (L138), as function (L177), Fe3Si (L228), etc. 

Please explain what the difference between the preliminary results reported in 15 and the results presented here are. 

Why do EDS on fracture surfaces - that is likely to be inaccurate - and not on polished specimens?

Figure 1 and others: It would make a lot of sense to show pristine sample as (a). 

It would be easier if you indicate exposure surface on all images. 

Figure 4, 8, 12: Maybe add background Na level in these images. It would be good if these images have the same axes scales for comparison. 
